# Partial cortico-hippocampectomy in cats, as therapy for refractory temporal epilepsy: A descriptive cadaveric study

Jessica Zilli[1]*, Monika Kressin[2], Anne Schänzer[3], Marian Kampschulte[4], Martin J. Schmidt[1]

1 Department of Veterinary Clinical Sciences, Small Animal Clinic, Justus-Liebig-University, Giessen, Germany, 2 Institute for Veterinary Anatomy, Histology and Embryology, Justus-Liebig-University, Giessen, Germany, 3 Institute of Neuropathology, Justus-Liebig-University, Giessen, Germany, 4 Department of Radiology, Justus-Liebig-University, Giessen, Germany

* Jessica.Zilli@vetmed.uni-giessen.de

**Data Availability Statement:** All relevant data are within the manuscript.

**Funding:** The author(s) received no specific funding for this work.

## Abstract

Cats, similar to humans, are known to be affected by hippocampal sclerosis (HS), potentially causing antiepileptic drug (AED) resistance. HS can occur as a consequence of chronic seizure activity, trauma, inflammation, or even as a primary disease. In humans, temporal lobe resection is the standardized therapy in patients with refractory temporal lobe epilepsy (TLE). The majority of TLE patients are seizure free after surgery. Therefore, the purpose of this prospective cadaveric study is to establish a surgical technique for hippocampal resection in cats as a treatment for AED resistant seizures. Ten cats of different head morphology were examined. Pre-surgical magnetic resonance imaging (MRI) and computed tomography (CT) studies of the animals' head were carried out to complete 3D reconstruction of the head, brain, and hippocampus. The resected hippocampal specimens and the brains were histologically examined for tissue injury adjacent to the hippocampus. The feasibility of the procedure, as well as the usability of the removed specimen for histopathological examination, was assessed. Moreover, a micro-CT (mCT) examination of the brain of two additional cats was performed in order to assess temporal vasculature as a reason for possible intraoperative complications. In all cats but one, the resection of the temporal cortex and the hippocampus were successful without any evidence of traumatic or vascular lesions in the surrounding neurovascular structures. In one cat, the presence of mechanical damage (a fissure) of the thalamic surface was evident in the histopathologic examination of the brain post-resection. All hippocampal fields and the dentate gyrus were identified in the majority of the cats via histological examination. The study describes a new surgical approach (partial temporal cortico-hippocampectomy) offering a potential treatment for cats with clinical and diagnostic evidence of temporal epilepsy which do not respond adequately to the medical therapy.

**Competing interests:** The authors have declared that no competing interests exist.

## Introduction

Epilepsy is a common neurologic disease in cats [1]. When animals experience epileptic seizures in absence of structural brain lesions, metabolic disturbances, or the influence of a toxin, they are diagnosed with idiopathic epilepsy (IE) [2]. In spite of evidence for genetic causes of IE in dogs [3], the association of specific genetic variants with feline epilepsy remains elusive [2]. However, great strides have been made in the evaluation of yet undetected epileptic diseases in cats formerly presumed to have IE. Although additional validation studies are necessary, important advances in neuropathologic investigation imply that the substrate of feline seizures can be a disorganization in the hippocampal cytoarchitecture and/or connectivity, as well as gliotic reactivity of the pyramidal cell band of the hippocampus fields (cornu ammonis [CA]-fields), referred to as hippocampal sclerosis (HS). These findings might be in part analogue to those described as a cause for human temporal lobe epilepsy (TLE) [4–8].

Different etiologies have been suggested to induce feline HS. Antibodies against voltage-gated potassium channel complex can induce limbic encephalitis and later degeneration of pyramidal neurons [9]. Such antibodies can be produced along underlying neoplasms [10]; however, in cats these antibodies were found mostly in absence of any neoplastic diseases [9]. A familial form of spontaneous epilepsy in cats has been described in association with genetically determined loss of neurons and gliosis in the hippocampus subfields [11]. Whereas HS may provide the primary basis of epileptogenesis in some cats [12], it has also been suggested that cats with chronic epilepsy may develop neuropathological changes consistent with HS as a consequence of seizures, often with a history of status epilepticus [4,13].

A presumptive diagnosis of HS can be made on living animals using magnetic resonance imaging (MRI) [14,15]. Hippocampal hyperintensities in T2-weighted and fluid-attenuated inversion recovery (FLAIR) images, as well as hippocampal contrast enhancement, seem to be significantly associated with HS [15]. Volumetric analysis of the hippocampus based on MRI images allows the determination of hippocampal atrophy, which can be another feature of HS [16,17]. In order to improve the detection of hippocampal pathology in MRI, optimized epilepsy protocols have been developed [18]. Research conducted to improve the diagnostic sensitivity of MRI for the detection of HS is of course not only motivated by the desire to rule out IE. Several investigations established an association of HS with resistance to antiepileptic drugs (AED) [4,9,13,19], which has a major influence on the individual prognosis of affected cats. As animals with insufficient seizure control are often euthanized, non-pharmacological treatment modalities are becoming increasingly important [20]. The resection of the hippocampus in cats with HS may be capable of improving their quality of life and seizure frequency, or even result in seizure freedom as it was shown for humans with TLE [21]. Human TLE is associated with AED resistance, and thus surgical resection of the hippocampus, rather than AED-based treatment, has become the therapy of choice and has thus far proved to have excellent clinical outcome [21,22]. Although hippocampectomy was suggested as a treatment option for resistant cats with HS [18,23], a surgical technique has not been introduced yet.

In this descriptive cadaveric study, we investigated the topographical morphology of the feline hippocampus using 3D brain and skull models to examine possible surgical approaches to the hippocampus in feline cadavers to allow removal of the hippocampus without harming the deeper brain structures, and to obtain standardized tissue specimen for histological evaluation.

## Materials and methods

### Cats

Fourteen cats of different age were collected from the Department of Veterinary Clinical Sciences, Clinic for Small Animals of the Justus-Liebig-University, Giessen. The cats were

euthanized or had died from diseases unrelated to the skull and central nervous system. Written consent was obtained from all owners that donated their animals for the study, and actual cats remained anonymous. Two cats were excluded due to the presence of cerebral disease in the pre-surgical MRI. Of the remaining twelve cats, ten were enrolled to perform the surgical procedure and the other two were used for the mCT study of vascular supply in the surgical field.

The cats were all but two (British Shorthair, Persian) Domestic Shorthair cats (DSH), weighing between 0.5 and 5.6 kg (median weight: 3.79 kg). Median age was 4.9 years (range: 1 month to 10 years). Of the twelve cats, six were females (3 spayed and 3 sexually intact) and six were neutered males (Table 1). Cadavers were eligible for the current study if they did not present morphologic brain abnormalities, as assessed through the pre-surgical MRI examination.

## MR- and CT-imaging of the head

A standardized MRI protocol was used to examine the brain before and directly after removal of the hippocampus using a 3.0-T superconductive system and a flex-mini sensitivity-encoding coil (Siemens Healthcare). Sagittal, dorsal, and transverse T2-weighted (TE = 120 ms, TR = 2.900 ms) and T1-weighted 3D sequences were acquired in all cats pre- and post-operatively. Slice thickness varied from 1–2 mm. The field of view measured 180 x 180 mm. The matrix was 288 x 288. Before surgery, CT data sets were also acquired with a 16-slice helical CT scanner (CT Brilliance, Philips, Hamburg, Germany; 120 kV, 350 mAs, matrix 512 x 512, slice thickness 0.8 mm, pitch 1).

## Image processing

In order to find the appropriate surgical approach to the hippocampus, 3D models of the head were built on the basis of MR- and CT images. Image processing for volume rendering of the

**Table 1. Signalment and cause of death of the study cats.**

| Animals | Breed | Weight (kg) | Age | Sex | Cause of death |
|---|---|---|---|---|---|
| 1 | Domestic Shorthair | 5.2 | 5 years | Male, neutered | Spinal fracture Th10 |
| 2 | Domestic Shorthair | 3.5 | 10 months | Male, neutered | Fractures of both femurs, fracture of the ileo-sacral joint on the right, fracture of the ileus, fracture of the 3rd coccygeal vertebra |
| 3 | Domestic Shorthair | 3.7 | 4 years | Male neutered | Suspected rupture of the trachea with pneumomediastinum |
| 4 | Domestic Shorthair | 3.5 | 9 years | Female, entire | Subcutaneous emphysema and pneumothorax due to rips fracture after trauma |
| 5 | Domestic Shorthair | 3.75 | 8 years | Male, neutered | Tarsal luxation and mandibular luxation |
| 6 | Domestic Shorthair | 2.7 | 10 years | Female, neutered | Inguinal wound and hypothermia after trauma |
| 7 | British Shorthair | 0.5 | 1 month | Female, entire | Atresia ani |
| 8 | Domestic Shorthair | 3.8 | 7 years | Female, neutered | Intestinal neoplasia |
| 9 | Domestic Shorthair | 4.2 | 5 years | Female, entire | Aortic thromboembolism |
| 10 | Domestic Shorthair | 5.6 | 9 years | Female, neutered | Aortic thromboembolism |
| 11 | Domestic Shorthair | 4.8 | 1 year | Male, neutered | High-grade mediastinal lymphoma |
| 12 | Domestic Shorthair | 4.2 | 11 months | Male, neutered | Pulmonary contusion and pneumothorax after trauma |

volumes of interest was achieved using specialized graphical software (AMIRA®, Mercury Computers Systems, Berlin, Germany). This program can combine image information of CT and MRI into 3D models. The hippocampus, brain surface, skull and skin surface were extracted from the digital images by manually tracking the boundaries of the structures as described previously [24]. Using landmark tools, the hippocampus was highlighted within the brain model and its position was projected on the surface of the brain and skull.

## mCT study

In order to visualize the blood vessels of the surgical area, the cerebral vasculature of two cats was perfused with Solutrast®. Directly after death the chest was opened, the brachiocephalic trunk was prepared, and a blunt cannula was inserted into the vessel. After tight fixation of the cannula and ligation of the vessel caudal to the cannula, the animals were first perfused with heparin (25000 UI) diluted in crystalloid solution (500 ml) to prevent the blood from clotting in the vessels. The blood in the cerebral vessels was drained out through an incision on the right auricle. Solutrast® was infused into the vascular system after complete drainage of blood. After exit of the Solutrast® from the right atrium, the chest was tamponed, and the carcass was stored at room temperature with the head hanging down for one night. Then, the brains were removed from the skull and fixed in formalin.

The brains of the perfused cats were examined with the mCT System Skyscan 1173 of the company Brucker Microct (Kontich, Belgium). The SKYSCAN 1173 is a high energy mCT scanner which includes a 130 kV microfocus (< 5 μm), X-ray source (Hamamatsu 130/300), flat panel sensor (detector), and a precision object manipulator. Before scanning, the brains were wrapped in parafilm in order to avoid dehydration and fixed into the sample holder with styrofoam. The sample holder was then positioned on the sample stage, between the X-ray source and detector, within the mCT.

The voltage and current source selected for the acquisition were respectively 70 kV and 114 μA. The detector had a matrix size of 2240x2240 pixel, with a pixel size of 20.1 μm. In order to reduce the beam hardening, a 1.00 mm aluminum filter was used. For each brain, two scans were necessary in order to obtain a complete 3D image. The samples moved with a rotation step of 0.25–0.30˚ on their vertical axis with a "step and shoot" type of motion, whereas the detector was still. The exposure time was between 950–1100 ms. The result of the acquisition process was raw data in TIFF format with a depth of 16 bits. Next, the raw data was reconstructed using a modified Feldkamp algorithm. As a result, three-dimensional data sets with isotropic voxels and 8-bit grey tone distribution were obtained and saved as BMP files. The digital processing and analysis of the 3D images was then performed with the program Analyze© 12.0, produced by the company Biomedical Imaging Resource (BIR) (Mayo clinic in Rochester, MN, USA). For the identification of the hippocampal vasculature and its isolation from the surrounding tissues, a grey threshold on a grey scale was set. Once the vessels of the hippocampus were extracted, they were inserted in the cerebral frame in order to allow better orientation. Relevant arteries in the surgical field were identified on the basis of published investigations [25–27].

## Tissue sample processing

All resected hippocampus specimens were fixed in 10% neutral-buffered formalin for at least 5 days at room temperature. Before paraffin embedding, the samples were taken off formalin and pre-treated with phosphate buffered saline (PBS), in order to wash out formalin. This process lasted 72 hours, and the PBS solution was replaced twice in order to optimize formalin diffusion out of the samples. The resected specimens were infiltrated with paraffin wax and then

embedded into wax blocks. These were then sectioned with a Leica RM2125RT Microtome. Seven μm thick sections were mounted on standard microscope slides. Before staining, the sections had to be deparaffinized through a passage in a 100% xylene solution and then hydrated. The slides were stained using Nissl staining (cresyl-violet). In order to obtain this staining, after the hydration process the sections had to be put in a 50% potassium-disulphite solution for 15–20 minutes before being washed with distilled water. Following this, the sections were ready for staining in a 1.5% cresyl-violet solution for 20 minutes at room temperature. After staining, a quick rinse in PBS was performed. The next step was a differentiation in alcohol: the slides were immersed first in a 70% ethanol solution, then twice in a 96% ethanol solution, and finally twice in a 100% ethanol solution to remove the excess stain. The dehydration process was then completed through a passage in xylene.

After resection of the hippocampus, all brains were fixed in formalin and processed in the same way as described above. 10–15 slides (76x52 mm) at different levels of the corticotomy were obtained and stained with the cresyl-violet staining. These slides were used to evaluate the extent of the resection and the presence of accidental traumatic injury in the structures mesial to the hippocampus.

## Results

### Localization of the hippocampus within brain and skull

Using transparent skull and brain models, the opaque hippocampus could be clearly visualized within the brain (Fig 1). The body and part of the tail of the hippocampus were identified underneath the caudal ectosylvian gyrus in mesocephalic as well as in brachycephalic cats (Figs 1 and 2). The most lateral point of the body of the hippocampus was localized at the level of the dorsal part of the caudal ectosylvian sulcus. Using landmark tools, the position of this part of the hippocampus was projected to the bone and skin surface (Fig 2). It was then localized using an imaginary line between the external occipital protuberance and the tip of the frontal process of the zygomatic bone. The middle of this line marked the center of the craniectomy site, approximately 0.5–1 cm rostral to the lateral cantus of the pinna.

Compared to mesocephalic cats, the brain of brachycephalic cats is distorted to accommodate their altered skull [24]. Due to the reduction in its longitudinal extension, the hippocampus is less curved but has the same position underneath the caudal ectosylvian gyrus. The concave side is slightly tilted towards the midline. The orbit of brachycephalic cats in this study was closed, and thus the tip of the frontal process could not be used as an orientation point. The external occipital protuberance and the base of the frontal process of the zygomatic bone, where it merges with the zygomatic process of the frontal bone can be used as a landmark in brachycephalic cats (Fig 2).

### Surgical technique

The cats were placed in sternal recumbency with the head positioned at a 45-degree rotation to the right. Silicone cushions were used to hold this position. Using the landmarks explained above, a straight dorso-ventral skin incision was made, from the sagittal crest to the temporo-mandibular joint (5–6 cm). The underlying fascia and the temporal muscle were dissected, and the two edges of the muscle were retracted using Weitlaner retractors to expose the parietal and temporal bone (Fig 3). After spreading the temporal muscle, the cranial sutures between the frontal, parietal, and sphenoid bone could be used as landmarks. The coronal suture (fronto-parietal suture) marks the rostral boundary, and the squamous suture (spheno-parietal suture) marks the ventral boundary of the craniectomy (Fig 3). A rectangular opening (ca. 2 x 3 cm) over the caudal sylvian and ectosylvian gyri was created 0.5–1 cm caudal to the

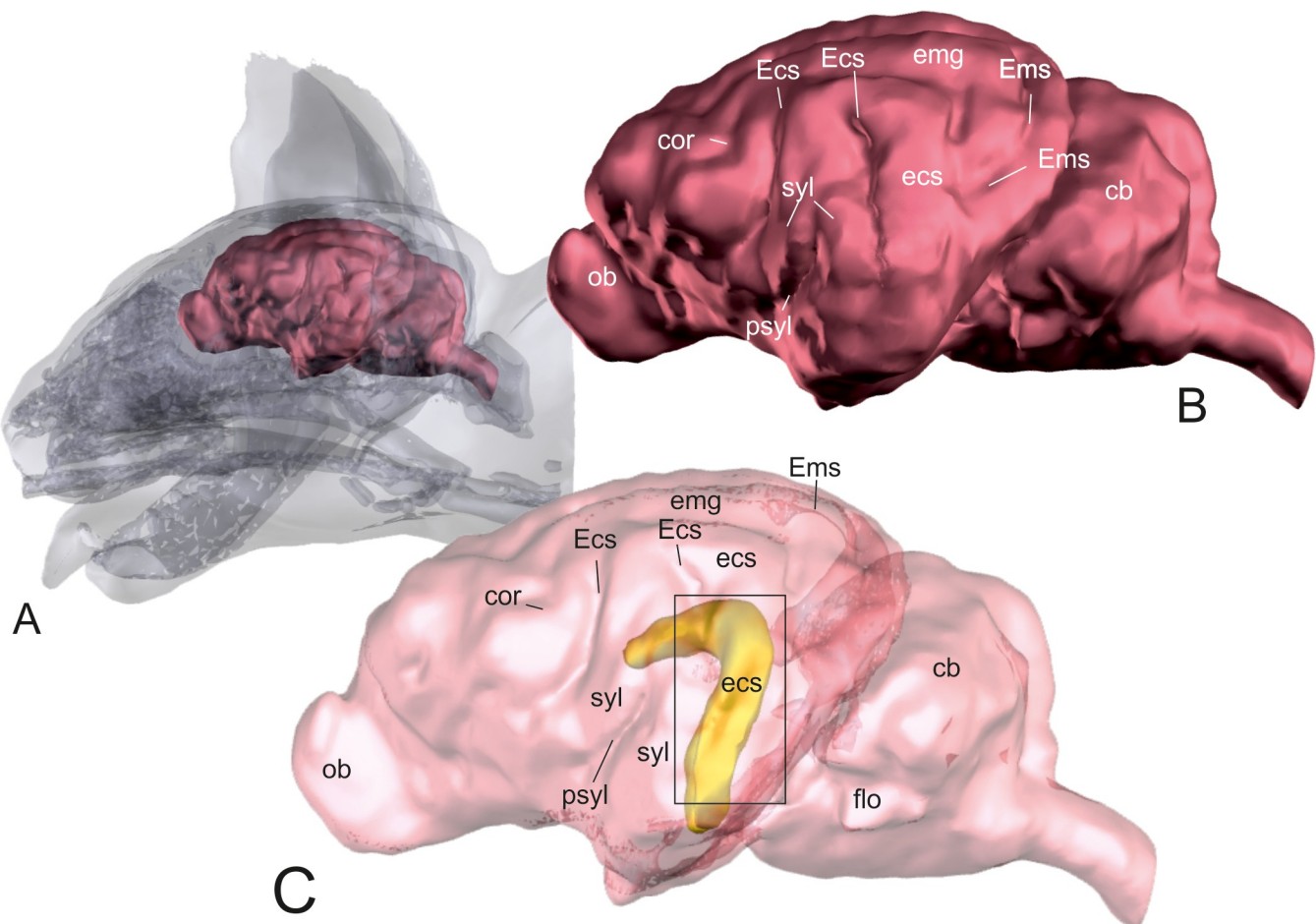

**Fig 1. Localization of the hippocampus within the brain.** 3D models of the skull and brain of a domestic shorthair cat based on CT- and MRI-images. Model A shows the whole brain (red) within the skull. Model B gives an overview of the neocortical sulci and gyri of the same brain. The brain surface is transparent in model C that displays the hippocampus (yellow) within the brain. Most of the hippocampus (head, body and parts of the tail) is situated underneath the caudal ectosylvian sulcus. ob: olfactory bulb; cor: coronal sulcus; Ecs: ectosylvian sulcus; ecs: ectosylvian gyrus; syl: sylvian gyrus; psyl: pseudosylvian fissure; emg: ectomarginal gyrus; Ems: ectomarginal sulcus; cb: cerebellum; flo: flocculonodular lobe.

coronal suture using a power-assisted drill and was enlarged as necessary using a Kerrison ron-geur (Fig 4A). The orientation on the cranial sutures worked well in both mesocephalic and brachycephalic cats. The underlying dura mater was opened with an upside-down Y-shaped incision using an 11-scalpel blade and Castroviejo scissors (Fig 4B). The large meningeal branches from the caudal and medial cerebral arteries were spared. The three parts of the dura could then be easily reflected to expose the caudal sylvian and caudal ectosylvian gyri (Fig 4C).

Unlike in other species, the caudal and rostral part of the ectosylvian sulcus does not have connective middle part in cats [28]. Thus, the upper end of the caudal ectosylvian sulcus was used as a leading structure for the incision into the cortical surface (Fig 4D). On the surface of the temporal lobe, at the level of the pseudosylvian fissure, the medial cerebral artery splits into three to four main branches [25]. From this point, the caudal main branch traverses the caudal sylvian gyrus and divides into subbranches, which run over the caudal sylvian and ectosylvian gyri. In some cats used in this study, the caudal ectosylvian gyrus also received vessels from branches of the caudal cerebral artery that approached from the dorsolateral aspect of the occipital lobe. These vessels were cauterized on both sides of the caudal ectosylvian gyrus. The

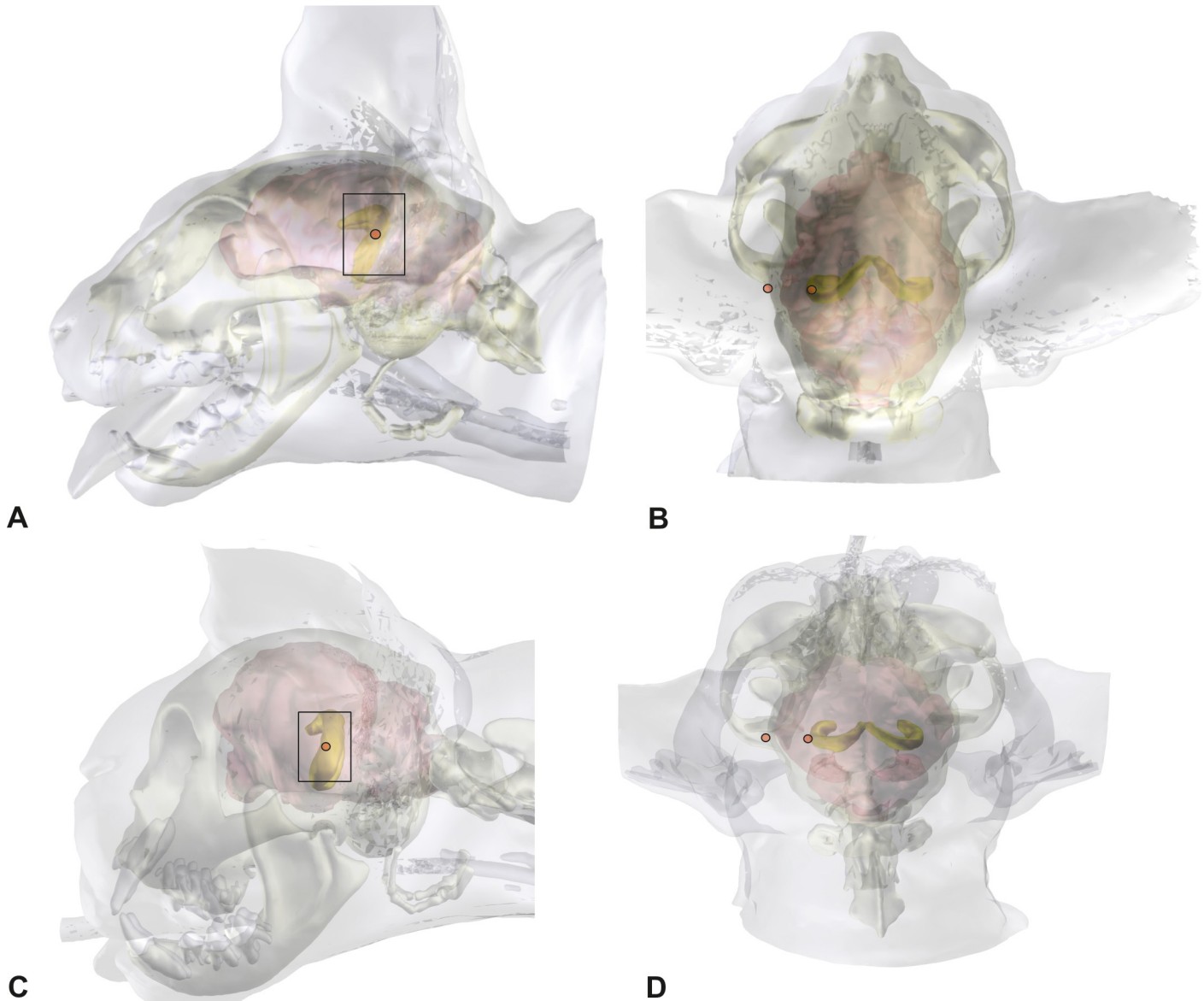

**Fig 2. Localization of the ectosylvian sulcus and the hippocampus from the skin surface and skull.** 3D models of the head, skull, and brain of a mesocephalic Domestic Shorthair cat (A,B) and a brachycephalic Persian cat (C,D), in lateral (A,C) and a dorsal view (B,D). The graphical software combines MRI- and CT-images. The hippocampus can be localized within the brain and the position can be projected on the brain and skull surface.

cortex of the ectosylvian gyrus was horizontally transected using an 11-scalpel blade 3–5 mm below the upper end of the ectosylvian sulcus (Fig 4D). The underlying white matter was gently dissected and gradually removed until cerebrospinal fluid emerged from the lateral ventricle. The alveus of the hippocampus contrasted with a glistening ebony color against the matt-white myelin of the surrounding cortical white matter (Fig 4E and 4F). The ependymal lining of the ventricle appeared greyish in color.

The opening in the pallium was ventrally enlarged on each side of the first cortical incision to further expose the hippocampus by gradually cutting the caudal ectosylvian gyrus left and right parallel to the adjacent sulci using the scalpel blade and Castroviejo scissors. The parenchyma of the ectosylvian gyrus was gently pulled laterally and ventrally using a spatula or a

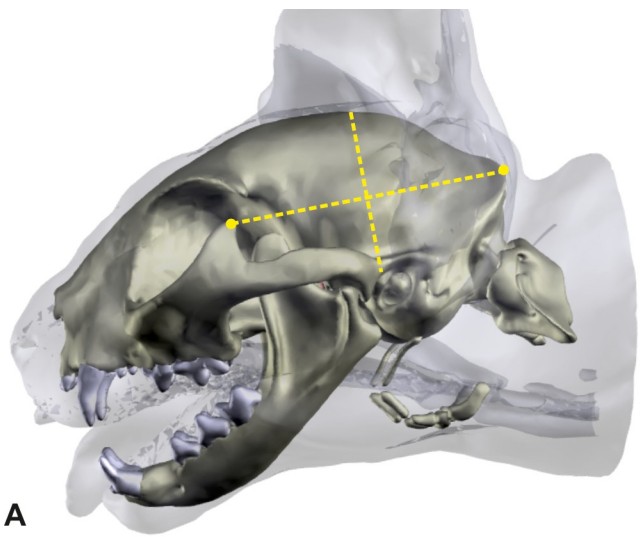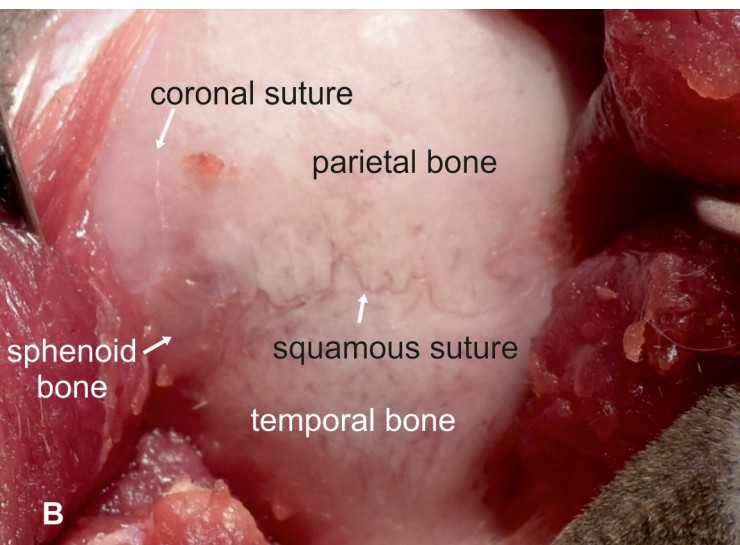

**Fig 3. Pterional craniotomy: Bony anatomical landmarks.** The cat's head is turned 45˚ to the side in order to place the zygomatic arch at the highest point. Landmarks for skin incision are the frontal process of the zygomatic bone and the external occipital protuberance. A skin incision is made in the middle of an imaginary line connecting the two landmarks (A). After dissection and spreading of the temporalis muscle, the left parietal and temporal bone is exposed. The crossing of the coronal and squamous suture is visible (B). The same landmarks that can be used for the orientation on the skin surface can be used at the calvaria as well. The middle point of these imaginary line marks the position of the craniectomy site with the hippocampus body at the center.

nerve hook and further dissected until the temporal horn of the ventricle was exposed (Fig 5A). The gyrus was cut on its ventral aspect and then fully removed. The cortical window could be mildly enlarged by reducing the adjacent white matter of the sylvian gyrus using surgical suction, exposing the transition from the hippocampus to the fimbria and the rostral choroidal artery, which was cauterized dorsally and ventrally on the exposed part of the hippocampus. A 1 x 2 cm neurosurgical patty (Neurosorb®) was moistened and placed at the ventral edge of the excised ectosylvian gyrus. A horizontal incision was made in the dorsal part of the exposed hippocampus and the cut was deepened and enlarged until the longitudinal hippocampal artery on the caudo-medial side was ruptured (Fig 5B). The procedure was repeated on the ventral aspect of the exposed hippocampus (Fig 5C). Next, the fimbria was transected. The isolated middle part was then gently pulled out by transecting its mesial connections with the help of a nerve hook and placed on the surgical patty (Fig 5D and 5E). The head and tail of the hippocampus were removed using surgical suction as much as possible under visual inspection.

At the end of the procedure, the dura mater was closed with a running suture using 5–0 Polyglactin. A standard closure method was then used to suture the temporal fascia, the subcutaneous tissues, and the skin. Operating time was approximately 120 minutes on average with this technique.

## Possible intraoperative complications

During the procedure, intraoperative complications were recorded. While opening the temporal fascia, the superficial temporal vein was damaged in one cat provoking hemorrhage. This vessel lies over the ventral border of the temporal muscle and should be avoided during this phase. Blunt dissection of the temporal musculature is preferable in order to prevent muscular bleeding and disruption, even if sharp dissection is faster and allows to prepare a wider bony surface. Excessive retraction of the temporal muscle rostrally could provoke a bulbus prolapse with traumatic damage to the optic nerve and secondary blindness can occur. While drilling the craniectomy, the transverse sinus was opened in one cat. For this reason, the craniectomy

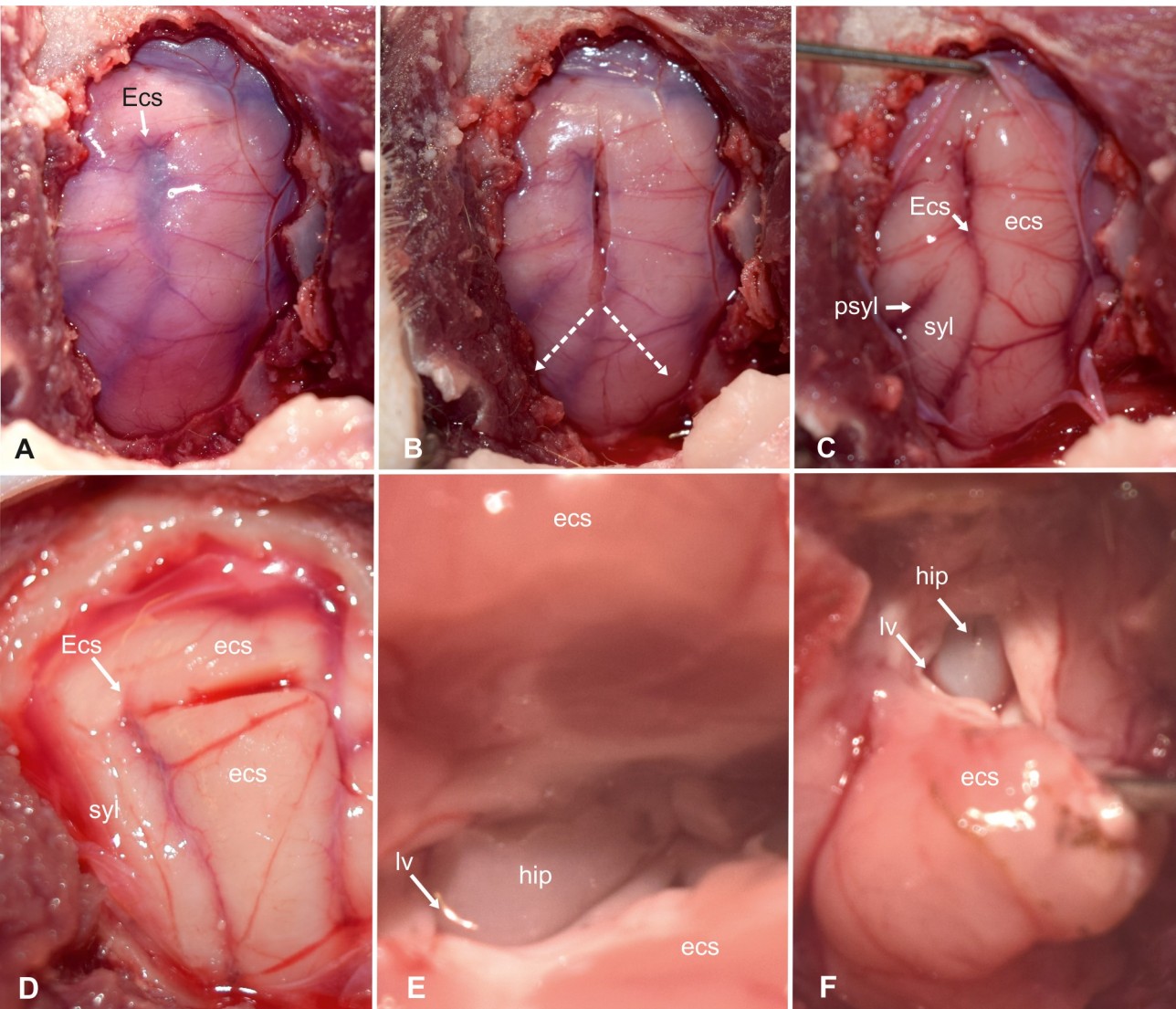

**Fig 4. Transcortical approach to the hippocampus (part 1).** Intraoperative photograph showing the view on the left sylvian and ectosylvian gyri and corresponding sulci after craniectomy and dural exposure (A). Tripartite dural incision (B). View of the brain surface after reflection of the dura (C). Magnified view on the incision in the dorsal ectosylvian gyrus (D). High magnification photograph of the transcortical access to temporal horn (E) and exposure of the hippocampal body (F). Ecs: ectosylvian sulcus; ecs: caudal ectosylvian gyrus; hip: hippocampus; lv: lateral ventricle syl: caudal sylvian gyrus.

should not reach the level of the nuchal crest. In all cats, the longitudinal hippocampal artery must be transected during the procedure which can cause considerable hemorrhage.

## Vascular supply of the hippocampus

mCT of the contrast-injected cat brains revealed the vascular supply of the hippocampal formation. The main blood supply to the hippocampus comes from the caudal cerebral artery which emits the longitudinal hippocampal artery at the medial base of the piriform lobe [25,26]. Both vessels follow the convex longitudinal axis of the medial side of the hippocampus (Fig 6). The large caudal cerebral artery runs more rostrally and curves around the medial geniculate body. The longitudinal hippocampal artery is much smaller. In all cats in this study,

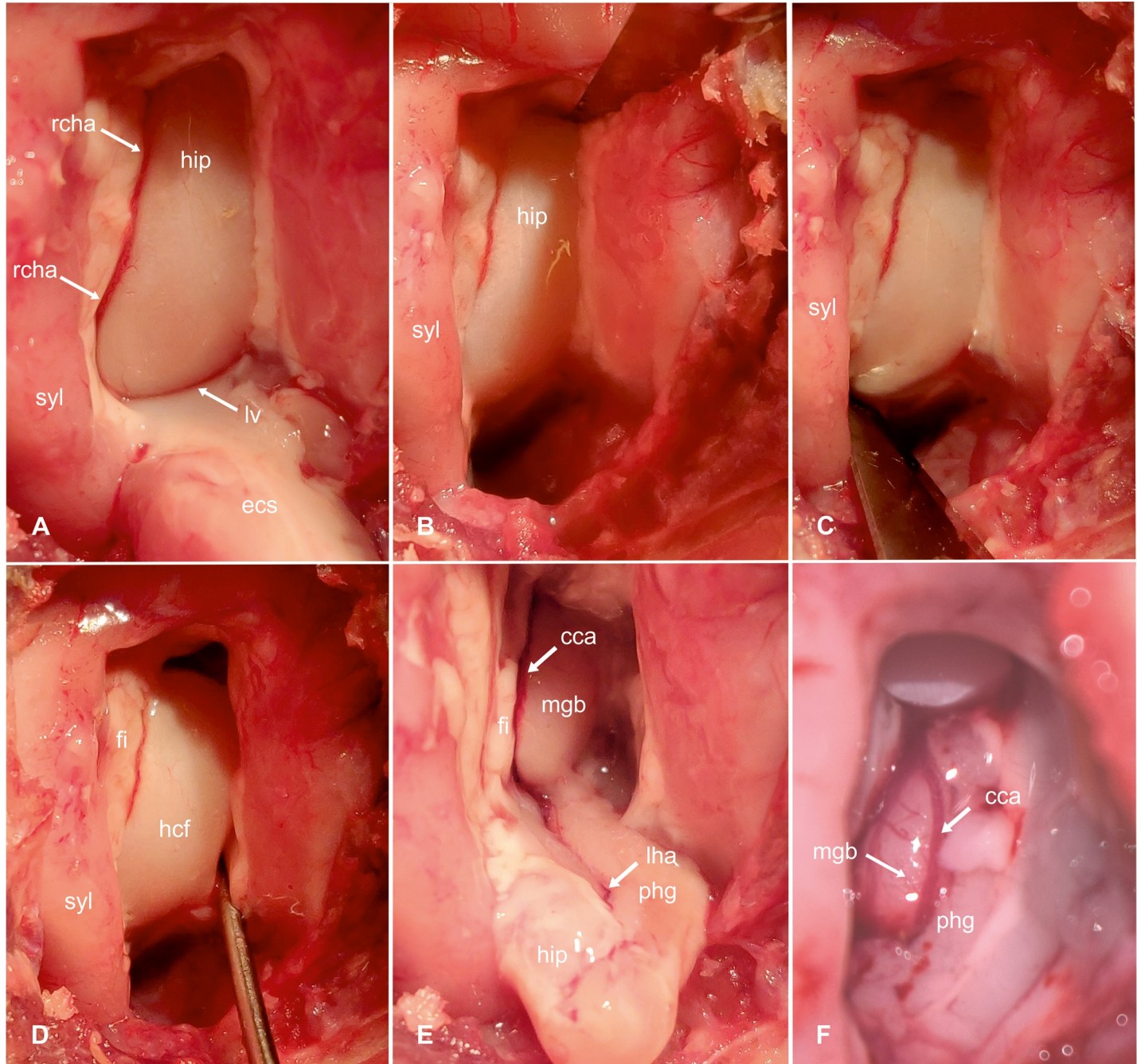

**Fig 5. Transcortical approach to the hippocampus-hippocampal resection (part 2).** Intraoperative photograph showing the hippocampus after resection of the temporal neocortex (caudal ectosylvian gyrus). The hippocampus is cut off dorsally (B) and ventrally (C) with an 11-scalpel blade; then its mesial connections are carefully transected with the help of a nerve hook (D) and the hippocampus is pulled out and removed (E). After completing the resection, the medial geniculate body and the caudal cerebral artery become visible (F, enlarged view). ecs: caudal ectosylvian gyrus; syl: caudal sylvian gyrus; rcha: rostral choroidal artery; hip: hippocampus; lv: lateral ventricle; fi: fimbria; hcf: hippocampal fissure; phg: parahippocampal gyrus; mgb: medial geniculate body; cca: caudal cerebral artery; lha: longitudinal hippocampal artery.

there was not a single artery, but rather this vessel was divided into two to three arteries sitting in the hippocampal fissure between the dentate and parahippocampal gyrus (Figs 5E and 6). From the longitudinal hippocampal artery, numerous segmental vessels run nearly parallel

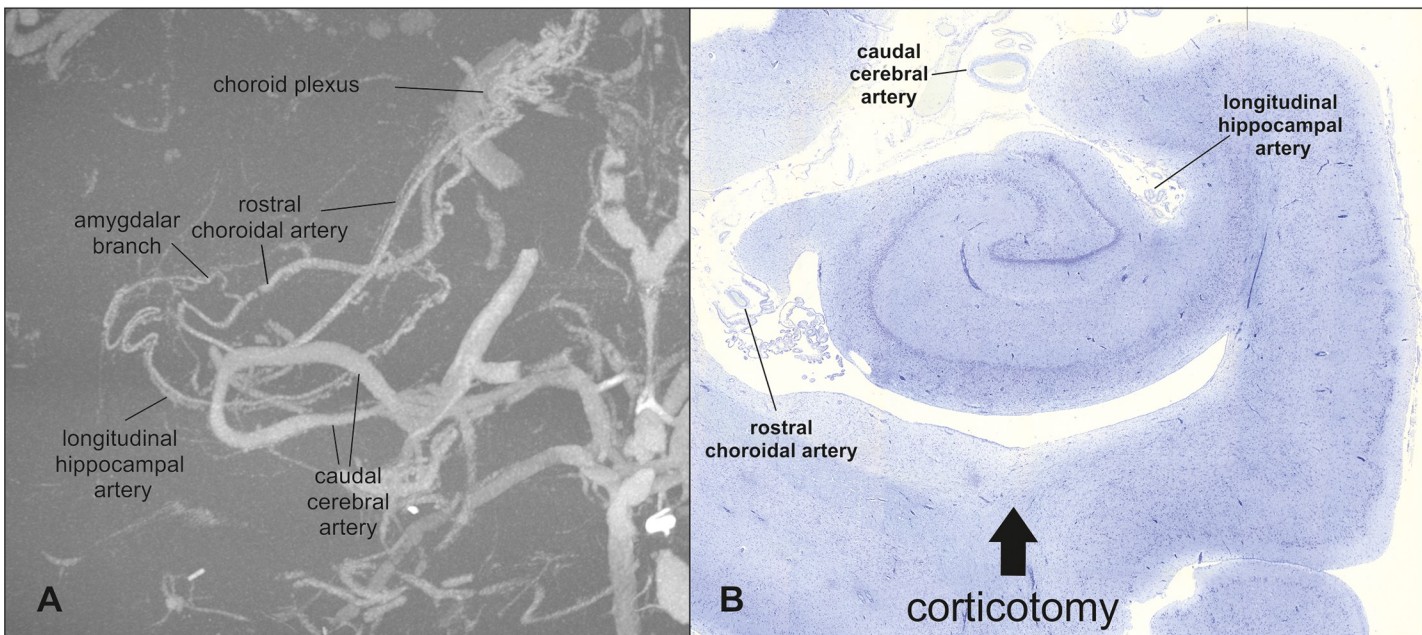

**Fig 6. Illustration of the blood supply of the hippocampus.** Maximum intensity projection of the contrast-injected vascular supply of the hippocampus in a dorsal view reveals the major blood vessels of the hippocampus (A). The brain and other vessel images have been removed using a grey threshold. The histopathological section of the hippocampus shows the relevant arteries in the surgical field (B).

towards the hippocampal fissure and supply the parenchyma of the hippocampus (as internal transverse hippocampal arteries) by penetrating the dentate gyrus and connecting with the rostral choroidal artery [25,26]. The rostral choroidal artery runs on the rostral side of the hippocampus in the shallow sulcus between fimbria and the hippocampus. This artery does not contribute to the hippocampal blood supply but gives rise into the choroid plexus of the third ventricle [25,26]. However, this artery must be considered as a source of hemorrhage during hippocampus resection.

### Postoperative MRI and histological evaluation of the resected tissue

Postoperative MRI and histopathological examination revealed that in all cats the whole body of the hippocampus could be resected en bloc (Fig 7A and 7B). In nine cats, there was no evidence of alterations in the surrounding structures. In one animal, the presence of a laceration in the thalamus was evidenced in the histopathological examination of the brain. The same lesion was not visible in the MRI post-resection. The entorhinal cortex and the amygdala adjacent to the hippocampus could not be removed.

All hippocampal specimens could be histologically examined and identification of the pyramidal and granular cell bands of all hippocampal fields was possible in five cats. In the other five, partial loss of these was evident. The subiculum, presubiculum and parasubiculum were in all cats damaged to a varying degree. In 3 specimens, the infrapyramidal blade of the dentate gyrus was also disrupted (Fig 7C–7E). The entorhinal cortex was not included in the specimen. The fimbria was in all cats torn off the hippocampus and was just partially visible.

### Discussion

Feline HS may occur as a primary disease of the hippocampus, or as a consequence of status epilepticus or chronic cluster seizures [4,11,29–31]. Regardless of whether hippocampal

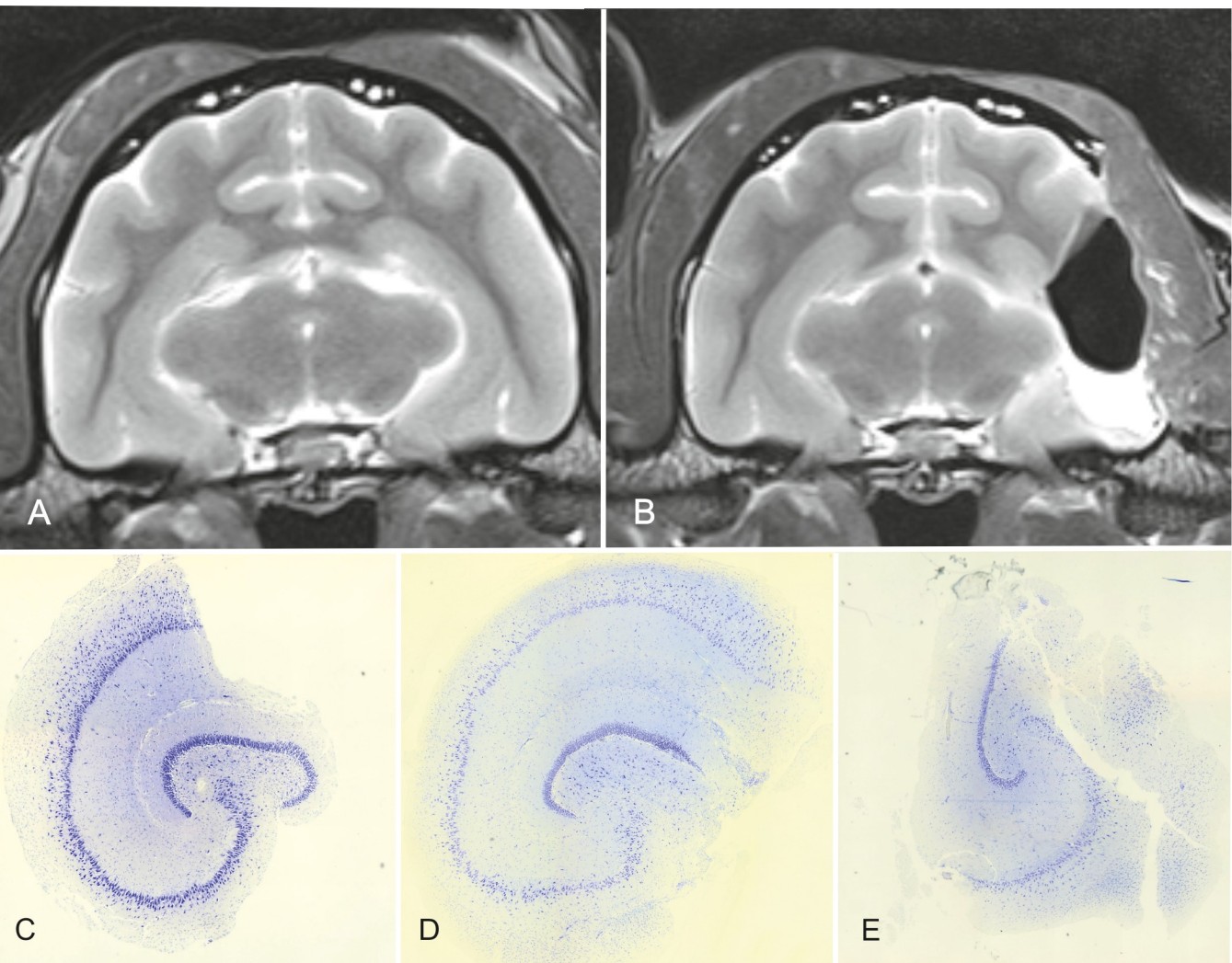

**Fig 7.** MRI before (A) and after (B) surgery and histological sections of the resected hippocampus (C-E). In the MRI post-resection, the left hippocampus (body) as well as the temporal neocortex are absent. Nevertheless, most of the hippocampal tail and the whole head are still in situ (B). Histological sections of the hippocampus after resection show that the specimen allow the examination of all CA-fields in most cases (C, D), but in some of them not all fields are preserved and can be examined (E). The completeness of the resected specimens is also related to the surgeon's learning curve.

sclerosis is cause or the consequence of seizures, the change in the cellular architecture is assumed to predispose for seizure development, propagation, and enhancement as well as for refraction to medication [4,18,20]. Surgical resection of the hippocampus and associated temporal lobe structures is considered an effective treatment for refractory seizures originating from HS in humans, and seizure freedom rates after temporal surgery range between 69–90% in humans [21,22]. Cats with seizures and HS may also benefit from hippocampal resection, which is why an evaluation of possible surgical techniques is the first step towards the establishment of systematic epilepsy surgery in animals.

Due to morphological differences, the standard technique described for humans cannot be directly adopted for cats. In humans, massive expansion of the temporal cortex caudal to the non-growing insula creates a structurally distinct temporal lobe that is somewhat separated from the rest of the hemisphere by the deep sylvian fissure [32]. The hippocampus head and body follow this curved movement during development and the bulk of the hippocampus and

amygdala lie separated from the rest of the hemisphere within the temporal lobe, which can be resected en bloc [33]. In cats and other domestic carnivores, the temporal lobe has not developed to a comparable extent. The feline hippocampus lacks a distinct tapering from a head into a tail [34,35]. The left and right hippocampi have a rather uniform shape in their dorso-ventral course around the thalamus. The tail sits far dorso-medially underneath the corpus callosum, and the head, associated with the amygdala, sits deep in the cerebral hemisphere adjacent to the thalamus [33,34,36]. The standard technique of anterior temporal lobectomy (also called cortico-amygdalohippocampectomy) is therefore not applicable for cats, as a structurally distinct temporal lobe has not evolved. Destruction of the hippocampus used to be frequently performed in dogs and cats in experimental studies. Aiming to understand the physiology of the hippocampus, researchers partially or totally removed the structure to study the effects of the lesions in living dogs and cats [37–42]. The main approach used in such studies was a dorsal paramedian craniectomy in the parietal bone. The hippocampus was destroyed inside the brain through a cortical resection window in the suprasylvian- and/or ectosylvian gyrus and aspirated using surgical suction [37–43]. Access to the hippocampus is limited in this approach and resection was incomplete in many animals. Furthermore, structural damage of the lateral and medial geniculate body occurred in some of the operated cats [44]. Although the cortical corridor may differ, the technique used in the old experimental studies resembles selective amygdalohippocampectomy in humans, in which hippocampus and amygdala are removed through different cortical corridors using an ultrasonic aspirator, a bipolar cautery, or suction [33]. If histological examination of the resected tissue is not desired, such a minimally invasive transcortical approach at variable sites might also be appropriate. In this case, the use of neuro-navigational systems and an intraoperative microscope are most likely necessary to carry out a successful surgery. In fact, thanks to these facilities, which are largely available in human neurosurgery, selective amygdalohippocampectomy, the less invasive approach to mesial temporal structures, can be performed in humans [33]. However, there is disagreement regarding as to whether this approach or the more invasive cortico-amygdalohippocampectomy also known as anterior temporal lobectomy is correlated with a better outcome and less post-operative complications [45–51]. In veterinary medicine, the use of these devices could also be of great help while performing mesial temporal resections, above all considering the aforementioned anatomical and size differences existing between human and feline brains [36,52–54].

However, we consider pathologic examination of the resected tissue of paramount importance to verify the presumptive diagnosis of HS and to clarify the role of HS in epileptogenesis and/or seizure perpetuation [4]. In this study, we designed an approach to remove the hippocampus and obtain specimens usable for histological examination at the same time. This approach is partly built upon existing craniectomy techniques that were slightly refined. To access most of the forebrain lobes, a standard rostro-tentorial craniotomy is described for dogs and cats [53]. Using the crossing of the squamous suture and the coronal suture as a landmark, opening of the skull in this approach resembles the pterional approach in humans. The Greek word "*pterion*" describes a localization on the lateral calvarium where the squamous suture meets the coronal- and spheno-squamosal suture. These sutures are also present on the skull of cats [34] and thus we suggest referring to the craniectomy used for hippocampectomy in cats as pterional craniectomy. A T-shaped incision of the temporal muscle was proposed to access the temporal lobe in dogs [55], but such an incision was not necessary to sufficiently expose the skull in the cats of this study. However, the tri-partite, or even better X-shaped, incision of the dura was vital to expose enough of the sylvian and ectosylvian gyri and to orientate in the surgical field. A simple straight incision did not allow sufficient exposure of all cortical landmarks. Particularly, the ventral end of a straight dural incision hindered preparation and removal of the caudal ectosylvian gyrus and placing of the surgical patty on which the

hippocampus bloc comes to lie after resection. Removal of the caudal ectosylvian gyrus was necessary to achieve resection of the hippocampus tissue en bloc. Spreading the cortical mantle after vertically cutting the caudal ectosylvian sulcus might also be possible to spare some of the structural integrity of the gyrus. However, spreading the ectosylvian gyrus to an extent that allows removal of the hippocampus en bloc could anyway lead to damaging the cortex in this gyrus and, more importantly, compressing the medial cerebral artery in the adjacent pseudo-sylvian fissure with devastating consequences for the vascular support of the hemisphere. Spreaders might also further limit approachability to the hippocampus.

In this study, the presence of a thalamic laceration possibly due to manipulation during surgery was evident in one cat. Since the same abnormality was not observed in the post-operative MRI, it was speculated that this could also be an artefact related to the histological preparation of the tissues. In fact, this is more likely to occur with less invasive approaches, in which the hippocampus cannot be directly visualized by the surgeon, in absence of neuronavigational systems as we know from the older studies [44]. With the approach described here it is indeed more likely to provoke ischemic damage or swelling of the surrounding structures due to vascular impairment and/or manipulation of the parenchyma.

The head and tail of the hippocampus could not be resected as a whole. However, as they were visible as a distinct structure, they could be removed using suction. Sclerosis of the hippocampus can occur in combination with neuronal loss and gliosis in the amygdala [11,56]. The epileptogenic role of the amygdala in cats has been emphasized in experimental studies [12,57,58]. Thus, additional resection of the amygdala could provide more effective seizure control than resection of the hippocampus alone [56]. The nuclear masses of the amygdaloid body lie further rostro-ventrally underneath the sylvian gyrus [34,35]. Removal of the sylvian gyrus in order to expose the amygdala might again carry the risk of damaging the medial cerebral artery. An approach to the amygdala might be possible by undermining the sylvian gyrus from the corticotomy site laterally, but the view into this area is extremely limited. Furthermore, the medial aspect of the amygdala lies in close proximity to the cavernous sinus. Removal of this brain section is hardly feasible using the approach described here.

The excised tissue did not include all parts of the hippocampal formation. The cutting line is through the subiculum and parahippocampal gyrus, which is why examination of these parts and the entorhinal cortex is not possible with this technique. The International League Against Epilepsy (ILAE) classifies HS into three subgroups based on neuronal loss and gliosis in the subfields of the hippocampus [59]. The subiculum and entorhinal cortex are not included in this classification system, and thus absence of these parts in the resected tissue seems to be acceptable at this time. A pioneering study investigating the hippocampus in a large cohort of cats with HS demonstrated that all CA fields can show cytopathological changes with the CA3 region being most frequently affected [4]. The quality of the specimen increased with increasing experience with the technique. In the end, almost all CA fields were well preserved in the tissue specimen and would therefore be sufficient to allow reproducible examination according to standards in human neuropathology [59]. Diagnosis of HS relies mostly on documentation of reduced pyramidal cell density and astrogliosis, which may differ throughout the hippocampal axis, also in healthy cats. If only the body of the hippocampus is available for examination, reference values on cellular density and cytoarchitecture for this part of the feline hippocampus would be necessary to establish standardized examination and diagnosis of feline HS.

The expected clinical manifestations of the loss of hippocampal tissue have been demonstrated in the resection studies of the past century. The caudal ectosylvian gyrus is part of the primary auditory cortex, which is tonotopically organized [60]. Unilateral loss of hearing frequencies between 2.5 and 9 mHz must be expected after resection of the caudal ectosylvian

gyrus, but there is no reason to expect total hearing loss on one side. Removal of the hippocampus was associated with deficits in motivation, spatial memory, and orienting response, as well as loss of conditioned reflexes and learned behaviors [38,42,61,62]. Cats can get hyperactive and aggressive [63] and may have a disruption of their diurnal cycle [38,44]. Moreover, due to the greater development of the temporal muscle in cats in comparison to humans, it can be speculated that cats could develop problems with chewing and food uptake after invasive dissection of this muscle. Nevertheless, no studies evaluating surgical procedures of the temporal area with involvement of the temporal muscle recorded such complications [64–67]. Furthermore, in our approach neither the muscular insertions nor its vascularization is compromised.

The known postoperative complications in human medicine can include fever, mild local pain, and mild to moderate headaches. These symptoms normally resolve over few days. In case they persist, it is indicated to perform a CT examination and a liquor study in order to rule out meningitis [22]. The development of a hematoma after resection is also a possible complication, but this can be removed in a second surgery [68]. In another study, the occurrences of infection of the bone lap (1.3%), mild hemiparesis (0.9%), hemianopia (0.4%), transient cranial nerve palsies (3.2%), transient postoperative language difficulties (3.7%), verbal memory deficits (8.8%), postoperative psychosis (2.3%) and postoperative depression (5.5%) are described as rare complications [21]. Global memory deficits occur rarely (1%), however after dominant hemisphere resection verbal memory deficits and language deficits happen very often (25–50% and 25–60%, respectively) [22].

Suitable candidates for surgical resection must be selected based on convergent lines of evidence implicating the hippocampus as epileptogenic region. Feline HS in association with feline partial seizures with orofacial involvement (FEPSO) seems to be a unique seizure disease because the structurally abnormal brain area (epileptogenic lesion), the area from which the abnormal electrical activity arises (epileptogenic zone), and the area that is responsible for the clinical manifestations of the seizures related to FEPSO (symptomatogenic zone) [23] are combined into a single structure [8]. Resective surgery might be most successful in these cats. However, a consensus about presurgical evaluation of a potential candidate for hippocampectomy will be necessary.

## Conclusions

Resection of the hippocampus and extraction of a hippocampus specimen for histopathological examination should be feasible in cats. Resected tissue does not include subiculum, entorhinal cortex, or amygdala. If seizures fail to respond to medication, then epilepsy surgery may be an option for cats with intravital diagnosis of HS.

## Acknowledgments

We wish to acknowledge the following individuals for their helping this study: Sigrid Kettner for her support in performing the histopathological examinations, Gunhild Martels for the mCT studies, and Ella Wenz for the MRI and CT studies.

## Author Contributions

**Conceptualization:** Martin J. Schmidt.

**Data curation:** Jessica Zilli, Martin J. Schmidt.

**Formal analysis:** Jessica Zilli.

**Investigation:** Jessica Zilli, Marian Kampschulte.

**Methodology:** Jessica Zilli, Monika Kressin, Marian Kampschulte.

**Project administration:** Monika Kressin, Anne Schänzer.

**Resources:** Monika Kressin, Martin J. Schmidt.

**Supervision:** Monika Kressin, Anne Schänzer, Martin J. Schmidt.

**Validation:** Anne Schänzer, Martin J. Schmidt.

**Writing – original draft:** Jessica Zilli, Martin J. Schmidt.

**Writing – review & editing:** Monika Kressin, Martin J. Schmidt.

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
