## [Decision Letter · Decision Letter 0]

22 Oct 2020

PONE-D-20-23336

Partial cortico-hippocampectomy in cats, as therapy for refractory temporal epilepsy: a descriptive cadaveric study

PLOS ONE

Dear Dr. Zilli,

Thank you for submitting your manuscript to PLOS ONE. After careful consideration, we feel that it has merit but does not fully meet PLOS ONE’s publication criteria as it currently stands. Therefore, we invite you to submit a revised version of the manuscript that addresses the points raised during the review process.

Please provide a detailed point by point answer to the Reviewer's concerns, on which I totally agree.

We look forward to receiving your revised manuscript.

Kind regards,

Tommaso Banzato

Academic Editor

PLOS ONE

Journal Requirements:

Reviewers' comments:

Reviewer's Responses to Questions

**Comments to the Author**

1. Is the manuscript technically sound, and do the data support the conclusions?

Reviewer #1: Yes

2. Has the statistical analysis been performed appropriately and rigorously? 

Reviewer #1: N/A

3. Have the authors made all data underlying the findings in their manuscript fully available?

Reviewer #1: Yes

4. Is the manuscript presented in an intelligible fashion and written in standard English?

Reviewer #1: Yes

5. Review Comments to the Author

Reviewer #1: The article of Zilli and Coll. describes a new and fairly accurate approach to partial cortico-temporal surgery for the treatment of refractory temporal epilepsy in cats.

The article is well written, well-documented and innovative. I have no objection to the rationale and core of the text.

However, I have some observation on the logical and practical application of the procedure:

• The AA repeatedly discuss the potential “damage” of surrounding brain areas induced by the proposed surgery. In my opinion this way to express the situation is wrong, as they are proposing and discussing a cadaver study. There is no “damage”, but a potential trespassing of the intended area defined as a target. I would rather use a different wording of their description. This is no simple whim, as I had to read the abstract and text accurately to understand the meaning they give to the “damage”.

• Matter of fact, the potential real damage induced by the procedure in living animals is probably greatly due to edema and inflammation of the surrounding tissue, and not by excessive removal of the target. Please comment

• Although they describe the external approach in some detail, in my opinion they may have underestimated the huge adverse effect of the surgery on the temporal muscle and related structure of the area. Considering the importance of the said muscle and temporo-mandibular joint in cats, I suggest to examine the potential negative consequence

• The AA described the experimental series of cats, that includes very small (young) individuals. A descriptive and detailed Table would help

• The AA. Should also cite the classic stereotaxic atlas of the cat brain by Snider et al. and later partial books on the same subject. Stereotactical surgery is way less invasive. I understand that the therapeutic approach envisioned here requires extensive tissue removal. However, the pros and cons of selective ablation or coagulation of limited brain areas focal to epilepsy induction should be at least briefly discussed.

6. PLOS authors have the option to publish the peer review history of their article (what does this mean?). If published, this will include your full peer review and any attached files.

Reviewer #1: **Yes: **Bruno Cozzi

---

## [Author Response · Author response to Decision Letter 0]

7 Dec 2020

Dear Mr Cozzi,

I would like to thank you for taking the time to reviewing our study. I very much appreciated your

comments and ameliorations and I have already integrated them into the paper. I would also use this

opportunity to respond to you and discuss each of the mentioned points.

Reviewer #1: The article of Zilli and Coll. describes a new and fairly accurate approach to partial

cortico-temporal surgery for the treatment of refractory temporal epilepsy in cats.

The article is well written, well-documented and innovative. I have no objection to the rationale and

core of the text.

Authors: I am extremely glad to hear that you like our approach for the resection of the

hippocampus in cats with antiepileptic drug resistant temporal epilepsy.

Reviewer #1: The AA repeatedly discuss the potential “damage” of surrounding brain areas induced

by the proposed surgery. In my opinion this way to express the situation is wrong, as they are

proposing and discussing a cadaver study. There is no “damage”, but a potential trespassing of the

intended area defined as a target. I would rather use a different wording of their description. This is

no simple whim, as I had to read the abstract and text accurately to understand the meaning they

give to the “damage”.

Matter of fact, the potential real damage induced by the procedure in living animals is probably

greatly due to edema and inflammation of the surrounding tissue, and not by excessive removal of

the target. Please comment

Authors:

We absolutely agree that the negative effects of tissue manipulation would be more profound in a

living animal and may rather include intraparenchymal haemorrhage and edema.

We would like to describe the potential complications of brain tissue manipulation while removing

the hippocampus. Therefore, we have tried to improve the wording in the text in order to support

the reader´s understanding of our concept of “damage”. Moreover, we added a paragraph in the

results including further complications we encountered during the study and that can occur also in

living animals.

After thoroughly studying the wording, we cannot see a semantic difference between primary

mechanical damage of the tissue and the secondary effects after mechanically

destroying/manipulating the tissue, which can also be described as damage, as general term. The

word injury has a synonymous meaning. Nevertheless, we made some specifications, hoping to

improve the comprehension of the text.

Reviewer #1: Although they describe the external approach in some detail, in my opinion they may

have underestimated the huge adverse effect of the surgery on the temporal muscle and related

structure of the area. Considering the importance of the said muscle and temporo-mandibular joint

in cats, I suggest to examine the potential negative consequence.

Authors:

Although we see the concern of the reviewer, the lateral transtemporal approach to the brain is a

standard technique in veterinary neurosurgery and the mechanical insult to the temporal muscle in

our approach seems to be relatively minor compared with the described basolateral approaches that

include resection of the zygomatic arch, of the condylar and/or coronoid processes of the mandibula.

We do not resect the muscle or widely detach its insertion or impair the vascular supply. The

temporo-mandibular joint remains unaffected. However, we have included a paragraph in the

discussion evaluating this concern.

Reviewer #1: The AA described the experimental series of cats, that includes very small (young)

individuals. A descriptive and detailed Table would help

Authors:

We have added the requested table in the section “Material and methods”.

Reviewer #1: The AA should also cite the classic stereotaxic atlas of the cat brain by Snider et al. and

later partial books on the same subject. Stereotactical surgery is way less invasive. I understand that

the therapeutic approach envisioned here requires extensive tissue removal. However, the pros and

cons of selective ablation or coagulation of limited brain areas focal to epilepsy induction should be

at least briefly discussed.

Authors:

We added the requested references, as well as a paragraph in the discussion reflecting on the use of

neuronavigational systems and its advantages and disadvantages. We are aware that a less invasive

approach would be possible if a histological examination is not required, nevertheless we consider it

as an extremely important point in patients with structural epilepsy. Indeed, also in human medicine

the examination of the resected specimen is always performed.

We hope you will appreciate the changes we apported to the paper and we look forward to your

reply.

Kind regards,

Jessica Zilli

---

## [Decision Letter · Decision Letter 1]

18 Dec 2020

Partial cortico-hippocampectomy in cats, as therapy for refractory temporal epilepsy: a descriptive cadaveric study

PONE-D-20-23336R1

Dear Dr. Zilli,

We’re pleased to inform you that your manuscript has been judged scientifically suitable for publication and will be formally accepted for publication once it meets all outstanding technical requirements.

Kind regards,

Tommaso Banzato

Academic Editor

PLOS ONE

Additional Editor Comments (optional):

Reviewers' comments:

Reviewer's Responses to Questions

**Comments to the Author**

1. If the authors have adequately addressed your comments raised in a previous round of review and you feel that this manuscript is now acceptable for publication, you may indicate that here to bypass the “Comments to the Author” section, enter your conflict of interest statement in the “Confidential to Editor” section, and submit your "Accept" recommendation.

Reviewer #1: All comments have been addressed

2. Is the manuscript technically sound, and do the data support the conclusions?

Reviewer #1: Yes

3. Has the statistical analysis been performed appropriately and rigorously? 

Reviewer #1: Yes

4. Have the authors made all data underlying the findings in their manuscript fully available?

Reviewer #1: Yes

5. Is the manuscript presented in an intelligible fashion and written in standard English?

Reviewer #1: Yes

6. Review Comments to the Author

Reviewer #1: The AA addressed all my comments and their answers are fully satisfactory, even when their opinion does not agree with mine

7. PLOS authors have the option to publish the peer review history of their article (what does this mean?). If published, this will include your full peer review and any attached files.

Reviewer #1: **Yes: **Bruno Cozzi

---

## [Editor Report · Acceptance letter]

4 Jan 2021

PONE-D-20-23336R1 

Partial cortico-hippocampectomy in cats, as therapy for refractory temporal epilepsy: a descriptive cadaveric study 

Dear Dr. Zilli:

I'm pleased to inform you that your manuscript has been deemed suitable for publication in PLOS ONE. Congratulations! Your manuscript is now with our production department. 

Kind regards, 

on behalf of

Dr. Tommaso Banzato 

Academic Editor

PLOS ONE